# Forensic Analysis of Polymeric Carpet Fibers Using Direct Analysis in Real Time Coupled to an AccuTOF™ Mass Spectrometer

**DOI:** 10.3390/polym13162687

**Published:** 2021-08-12

**Authors:** Torki A. Zughaibi, Robert R. Steiner

**Affiliations:** 1Department of Medical Laboratory Technology, Faculty of Applied Medical Sciences, King Abdulaziz University, P.O. Box 80216, Jeddah 21589, Saudi Arabia; 2King Fahad Medical Research Center, King Abdulaziz University, P.O. Box 80216, Jeddah 21589, Saudi Arabia; 3Virginia Department of Forensic Science, Richmond, VA 23219-1416, USA; robert.steiner@dfs.virginia.gov

**Keywords:** polymer, carpet fiber, direct analysis in real time, time of flight, mass spectrometry, function switching, oleamide

## Abstract

Polymeric fibers are encountered in numerous forensic circumstances. This study focused on polymeric carpet fibers most encountered at a crime scene, which are nylons, polyesters and olefins. Analysis of the multiple polymer types was done using Direct Analysis in Real Time (DART^™^) coupled to an Accurate time-of-flight (AccuTOF™) mass spectrometer (MS). A DART gas temperature of 275 °C was determined as optimal. Twelve olefin, polyester, and nylon polymer standards were used for parameter optimization for the carpet fiber analysis. A successful identification and differentiation of all twelve polymer standards was completed using the DART-AccuTOF^™^. Thirty-two carpet samples of both known and unknown fiber composition were collected and subsequently analyzed. All samples with known fiber compositions were correctly identified by class. All of the remaining carpet samples with no known composition information were correctly identified by confirmation using Fourier-transform infrared spectroscopy (FTIR). The method was also capable of identifying sub-classes of nylon carpet fibers. The results exhibit the capability of DART-AccuTOF^™^ being applied as an addition to the sequence of tests conducted to analyze carpet fibers in a forensic laboratory.

## 1. Introduction

Unlike DNA and fingerprints, fibers are associative evidence in that they establish a link between a person and a crime scene, object, or another person. Fibers are encountered as evidence in many different crimes such as hit and run, breaking, and entering, and rape. This exchange of material is a result of Locard’s exchange principle, which states that when two objects are in contact with one another, a transfer of material will occur.

Nylons, also referred to as polyamides, are a group of synthetic polymers that are used to manufacture rope, fabric, carpet fibers, and more [1] Various kinds of nylons are differentiated based on their synthesis [2]. A few examples of the various nylons include nylons 6, 6/6, 11, etc. The number(s) following the nylon indicates the number of carbon atoms that originated from the dicarboxylic acid and diamine groups that form the different molecular structures [3].

Polyesters are also synthetically produced polymers used for apparel, textiles, and carpeting. In terms of amount produced and shipped to customers, polyesters rank second only to cotton. There are many different polyesters such as polyethylene terephthalate (PET), polybutylene terephthalate (PBT), and Poly (1,4,-cyclohexylenedimethylene terephthalate), (PCDT). PET is found more abundantly because it is the polymer of choice in over 95% of all polyester fibers manufactured [4,5]. Another kind of polyester, olefin fibers are known for to be tough fibers that have high resistance to abrasion, thus affording good application in carpet fibers [6]. Ethylene and propylene are the only two saturated hydrocarbons that serve to synthesize polyolefin for fibers [5].

The researchers have applied DART to a wide selection of problems of interest to forensic analytical chemistry, but the “maturity” of DART varies considerably across the field. DART-MS can analyze surfaces without extraction steps and without damaging the surface; it can be used to study substrates not ordinarily amenable to analytical methods, including plants and plant-derived materials. In 2011, Adams published a study using DART-MS to analyze paper, noting that the chemical composition of the pulp and additives present in paper can help to determine its age, aiding in determining its historical authenticity [7]. To analyze the chemical composition of fibers, Direct Analysis in Real Time (DART™) coupled to an accurate time-of-flight (AccuTOF™) mass spectrometer (MS) was utilized.

This involves an ambient ionization ion source which requires little-to-no sample preparation [8,9]. DART has been authorized for analyzing many types of samples encountered at a forensic lab ranging from accelerants and explosives, to inks and controlled substances [10,11,12,13]. The ability of DART to test solids, liquids, and gases (via headspace) with no prior sample preparation makes this technique ideal for the analysis of carpet fibers. Forensic analysis of carpet fibers involves using microscopic examinations, a sequence of microchemical tests, and either Fourier-transform infrared spectroscopy (FTIR) or pyrolysis-gas chromatography (pyrolysis-GC). While FTIR is capable of identifying polymer subtypes, it involves a more tedious sample preparation step when compared to this method. Whereas with the destructive techniques using pyrolysis-GC and microchemical tests, their main disadvantage compared to the method described in this study is they both take more time for analysis. With the DART, several samples, including a calibration standard, can be identified and analyzed within minutes. In this study, the capability of the DART-AccuTOF^™^ was tested to analyze carpet fiber samples by distinguishing the various polymer types and sub-types based on their chemical composition.

The novelty of the work lies in the DART-AccuTOF™ being capable of identifying the various polymer types in carpet fibers and could be implemented as one of the many series of tests conducted when analyzing fiber evidence to increase the power of discrimination. It could serve as a quick screening tool, especially in cases with many fibers.

## 2. Materials and Methods

### 2.1. Materials

The polyester standards used were PBT Poly (1, 4-butylene terephthalate) (Molecular weight of repeat unit: 220.23 g/mol.), PCDT Poly (1,4-cyclohexanedimethylene terephthalate) (Molecular weight of repeat unit: 226.32 g/mol.), and PET Poly (ethylene terephthalate) (Molecular weight of repeat unit: 192.2 g/mol) were obtained from Scientific Polymer Products, Inc. (Ontario, NY, USA) in a polymer sample kit. The olefin standards were isotactic polypropylene and polyethylene was purchased from Sigma-Aldrich (St. Louis, MO, USA). All standards were stored in glass vials and in pellet form (with the exception of polypropylene). Polypropylene was in a powder form, which was melted and allowed to solidify to facilitate sampling.

#### Characterization

Attenuated total reflection spectra in the range 4000–400 cm^−1^ of the polymers were measured with Fourier transform infrared (FTIR) spectroscopy (Nexus 8700, Thermo Nicolet, Madison, WI, USA) and Omnic software version number 9.1.26 (Thermo Fisher Scientific Inc., Waltham, MA, USA) and a uniform resolution of 2 cm^−1^ was maintained in case of polymers. Analysis was performed on a DART (IonSense, Inc. Saugus, MA, USA) ion source coupled to a JMS-T100LC AccuTOF™ (JEOL Inc., Peabody, MA, USA) mass spectrometer using modifications of a previously published method [14]. In brief, 2 mg/mL polyethylene glycol in methanol solution (PEG 600), was used for exact mass calibration. Calibration was done by dipping the sealed end of a capillary tube (Kimble Glass Company, Vineland, NJ, USA) in the PEG 600 calibration solution and “wanding” the capillary tube in the sample gap for a few seconds. JEOL MassCenter software version 1.3.4 m (JEOL Inc., Peabody, MA, USA) was used to gather and analyze the data. Each data file contained a calibration curve developed from the PEG 600 calibration standard and was subsequently applied to all polymer data collected.

### 2.2. Experimental

Thirty-two carpet samples of various blends and colors were collected from local hardware stores. Each sample was packaged separately in a plastic bag and labeled (Sample 1–32). Of the thirty-two samples, fifteen had their fiber compositions included on the label. The remaining seventeen samples were treated as unknowns. Of the fifteen known samples, only three specific fiber types were observed: nylon, polyester, and olefin. It is also common to encounter carpets that contain a blend of several of these fiber types.

To obtain optimum results, parameters were first established for the DART-AccuTOF for the different polymer standards. The major parameters of importance were (1) the temperature of the DART gas stream and (2) the orifice 1 voltage of the AccuTOF™. An initial range of 200–300 °C gas temperature was chosen because of the melting points of the polymers shown in Table 1.

## 3. Results and Discussion

### 3.1. FTIR Spectroscopy

#### 3.1.1. Nylon 6/6

IR spectroscopy has been usefully applied for identification of the basic structural units present in the chemical configuration of nylon-66. The FTIR spectra of the nylon-66 is presented in Figure 1. The complete vibrational band assignment is made available for the selected polymeric materials, thereby confirming their molecular structure. The vibrational frequencies of all the fundamental bands and probable assignments are given in Table 2. The primary motivation for determining the molecular structure of a polymer using FTIR spectroscopy is to confirm its presence in the carpet samples.

#### 3.1.2. Polyester

The peaks in the IR spectra of the polyester carpet fabric are shown in Figure 2 appeared in the range of 600–4000 cm^−1^. The 1715 cm^−1^ shows C=O stretching vibration, 1409 cm^−1^ is attributed to aromatic ring, 1331 cm^−1^ and 1021 cm^−1^ shows carboxylic ester or anhydride, 1021 cm^−1^ indicates the presence of O=C–O–C or secondary alcohol, 967 cm^−1^ is of C=C stretching, and the 869 cm^−1^ peak shows five substituted H in benzene. The main structure of the polyester sample had ester, alcohol, anhydride, aromatic ring, and heterocyclic aromatic rings.

#### 3.1.3. Olefins

The major peaks for the olefins include C=C stretch from 1680–1640 cm^−1^, =C–H stretch from 3100–3000 cm^−1^, and =C–H bend from 1000–650 cm^−1^.

The IR spectrum of olefins is shown in Figure 3. The band greater than 3000 cm^−1^ is attributed to the =C–H stretch and the several bands lower than 3000 cm^−1^ for –C–H stretch. The C = C stretch band is attributed to 1644 cm^−1^. Bands for C–H scissoring (1465) and methyl rock (1378) are marked on this spectrum; in routine IR analysis, these bands are not specific to an alkene and are generally not noted because they are present in almost all organic molecules (and they are in the fingerprint region). The bands at 917 cm^−1^ and 1044 cm^−1^ are attributed to =C-H bends.

### 3.2. Mass Spectra of Polymeric Samples

Parameters were selected to try and identify a temperature in which all polymer types could be identified. It was established that a helium gas temperature of 275 °C yielded the greatest compromise of response for the various polymer types. Figure 4 and Figure 5 illustrate the total ion responses at temperatures ascending or descending in 25 °C intervals between 200–300 °C for PET and polypropylene, respectively.

#### 3.2.1. Nylon

When referring to Table 3, the protonated monomer [M+H] of nylon 6/6 is equal in weight to the nylon 6 protonated dimer [2M+H]. One potential cause for concern would have been the fragmentation of the nylon 6/6 monomer [M+H] into a 114.0919 *m*/*z* peak (equivalent to the nylon 6 monomer [M+H]). However, the protonated nylon 6/6 monomer [M+H] does not produce a fragment ion to a *m*/*z* equivalent to the nylon 6 protonated monomer [M+H], but rather into 100.1140 *m*/*z* that increases in intensity as the voltage of orifice 1 increases. In addition, as the orifice 1 voltage increases, the nylon 6 protonated dimer [2M+H] and trimer [3M+H] decreases in intensity, with the fragmentation resulting in a more abundant monomer [M+H]. While both the protonated monomer [M+H] and protonated dimer [2M+H] peaks of nylon 6/6 decrease in their abundances, a fragment ion is produced at *m*/*z* 100.1140, which in turn increases specificity among these similar yet different nylon types.

Table 3 shows the differences in the exact masses for the different monomers of PBT, PCDT, and PET. The difference in mass spectra between the three polyester types is shown in Figure 6.

#### 3.2.2. Olefin

The two olefins of interest proved to be particularly problematic in terms of identification. Because both polyethylene (C_2_H_4_) and polypropylene (C_3_H_6_) share the same empirical formula (C_n_H_2n_), many of their various units will have the same exact mass. Similar to the case with nylon 6 and 6/6, the protonated trimer of polyethylene has the same mass as the protonated dimer of polypropylene (*m*/*z* 85.1017). In the spectra of the two standards, an obvious difference can be seen (Figure 7).

In the polyethylene spectrum, peaks from the polymer backbone are observed ranging from approximately *m*/*z* 220–450. Each of those peaks was correctly identified as the different polyethylene monomers using the SearchFromList software. A difference of approximately 28 Da was observed between the repeat units of the polymer backbone, which is the mass of the polyethylene monomer [M] (C_2_H_4_ = 28 Da). However, with polypropylene, an expected difference of 42 Da (C_3_H_6_) between succeeding peaks was not observed. Although a noticeable difference in the spectra was observed, the majority of the peaks identified using SearchFromList were those that were common between both olefin types and thus could not be correctly identified as a polypropylene.

An interesting phenomenon occurred with the polypropylene standard. It is best explained when looking at the lower mass range in Figure 8. The differences observed between the “triplets” of peaks are a difference of 14 Da, which could theoretically be a methyl group. This is most likely due to the isotactic nature of the polypropylene standard. The *m*/*z* 85 peak is attributed to the protonated dimer [2M+H] of polypropylene [C_6_H_12_ + H], and the *m*/*z* 99 peak is the protonated dimer of polypropylene plus a methyl group [C_6_H_12_ + CH_2_ + H]. Due to slightly different end group terminations, other protonated molecules are seen superimposed with the polypropylene dimer and other mers (Figure 8).

### 3.3. Carpet Samples

Following the successful differentiation of the various polymer standards, the carpet fibers were tested. Of the 15 known carpet samples, there were two labeled as 100% nylon carpet samples (#19,23), six labeled as 100% polyester carpet samples (#5,20,24,28–30), and four labeled as 100% olefin samples (#21,22,27,32). The remaining known samples (#25,26, and 31) were labeled as mixtures of 91% olefin and 9% nylon, 91% olefin and 9% nylon, and 80% polyester and 20% nylon, respectively.

All known nylon, polyester, and olefin carpet samples were correctly identified using the DART-AccuTOF^TM^. Figure 9 shows representative spectra of the various fiber types.

Following the parameter optimization of the DART-AccuTOF^™^ and the sampling of the standards and known carpet fibers, it was determined whether this technique could compliment the FTIR results for unknown carpet fibers.

The DART-AccuTOF^™^ correctly identified all the remaining unknown samples as their corresponding polymer type. It was even able to distinguish the sub-type for two of the known nylon samples, where sample 3 was identified as nylon 6. Sample 4 was identified as nylon 6/6 as the latter did not produce a peak at *m*/*z* 114 (nylon 6 monomer [M+H]) [14]. The results were confirmed via a subsequent microchemical test using a few drops of 15.5% HCl, which instantaneously dissolves nylon 6 as opposed to other nylon sub-types as described in the author’s previous publication [14]. The results of the micro solubility tests helped confirm that the DART-AccuTOF can differentiate the different polymer sub-types in addition to identifying the basic polymer types.

## 4. Conclusions

The carpet samples with no known composition information were correctly identified by using Fourier-transform infrared spectroscopy (FTIR). The method was also capable of identifying sub-classes of polyester, olefin, and nylon carpet fibers.

DART-AccuTOF effectively differentiated numerous polymer types and sub-types based on the polymer chemistry of their monomers [M+H], and their associated dimers [2M+H] and trimers [3M+H]. The DART gas stream temperature of 275 °C was used as a compromise due to the differences in melting points; however, data collected at that temperature for all polymer types produced acceptable results. This technique can be applied in the testing of other fabric fibers, as well as objects that are made using these polymers.

DART-AccuTOF™ successfully identified the carpet sample’s polymer class (nylon vs. polyester vs. olefin) and successfully differentiated a nylon 6 carpet sample from a nylon 6/6. Due to the lack of other polyester carpet types, PET, which accounts for over 95% of the polyester production as previously stated, was identified in all the polyester carpet samples. In all the PET carpet samples, the *m*/*z* 204 ion peak was observed and therefore can serve as an aid in identifying the sample as a PET carpet sample. The lack of a PCDT or PBT carpet sample limits the conclusion that can be drawn with regard to the *m*/*z* 204 ion peak and polyester carpet samples. The same can be said about olefins, a differentiation of the olefin type could not be determined using DART-AccuTOF™. The presence of the *m*/*z* 85 ion peak and the polymer backbone will help assist with identifying the sample as an olefin. Additionally, due to the lack of known polypropylene or polyethylene carpet samples, a differentiating criterion could not be established.

In conclusion, DART-AccuTOF™ is capable of identifying the various polymer types in carpet fibers and could be implemented as one of the many series of tests conducted when analyzing fiber evidence to increase the power of discrimination. It could serve as a quick screening tool, especially in cases with many fibers.

## Figures and Tables

**Figure 1 polymers-13-02687-f001:**
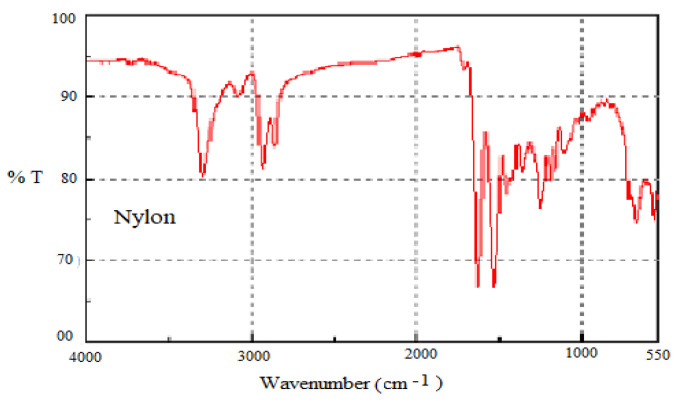
FTIR results for the Nylon 6/6.

**Figure 2 polymers-13-02687-f002:**
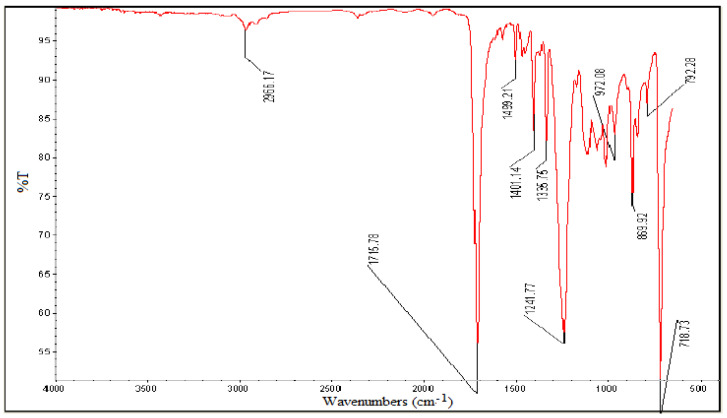
FTIR of Polyester.

**Figure 3 polymers-13-02687-f003:**
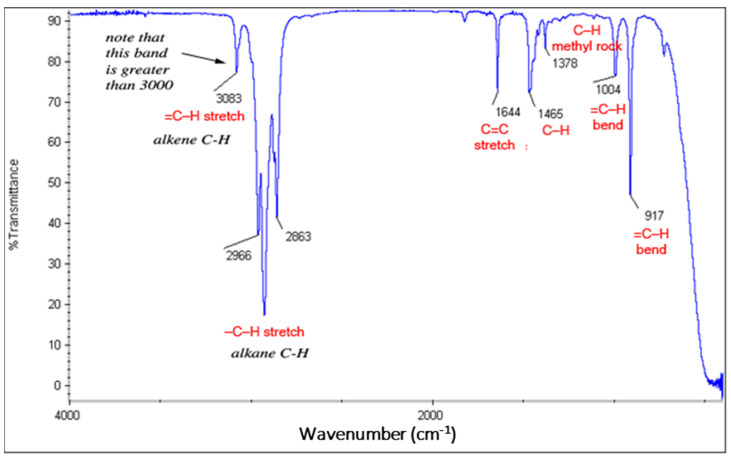
FTIR of Olefin.

**Figure 4 polymers-13-02687-f004:**
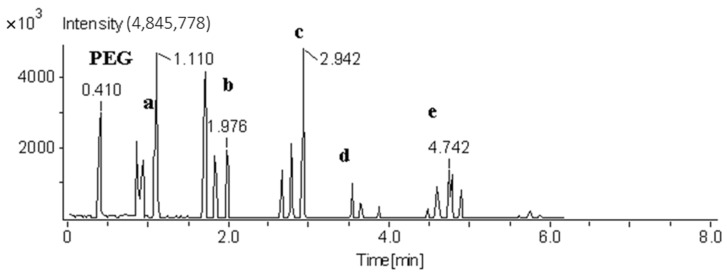
Total ion response of PET at temperatures: (**a**) 300 °C, (**b**) 275 °C, (**c**) 250 °C, (**d**) 225 °C, and (**e**) 200 °C.

**Figure 5 polymers-13-02687-f005:**
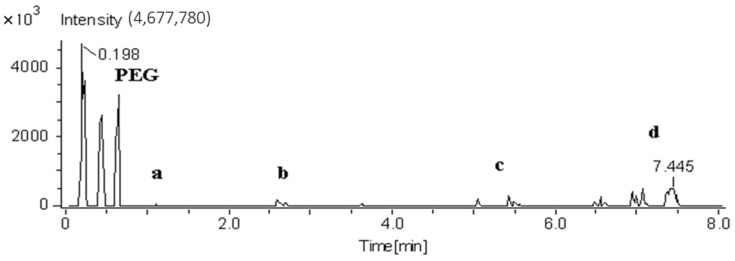
Total ion response of polypropylene at temperatures: (**a**) 200 °C, (**b**) 225 °C, (**c**) 250 °C, (**d**) 275 °C.

**Figure 6 polymers-13-02687-f006:**
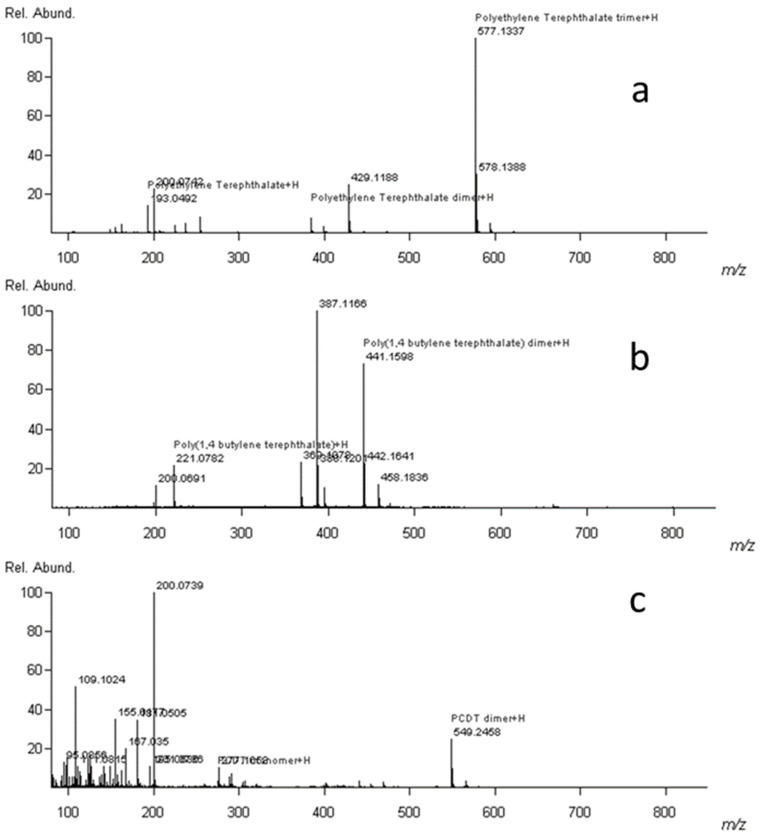
DART-AccuTOF^™^ mass spectra for standards (**a**) PET, (**b**) PBT, and (**c**) PCDT at 30 volts with correct peaks identified using SearchFromList software.

**Figure 7 polymers-13-02687-f007:**
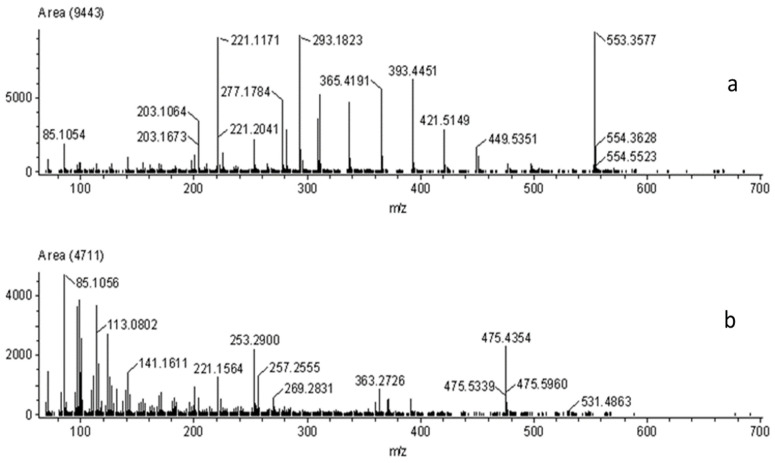
DART-AccuTOF^™^ mass spectra for standards (**a**) polyethylene and (**b**) polypropylene at 30 V.

**Figure 8 polymers-13-02687-f008:**
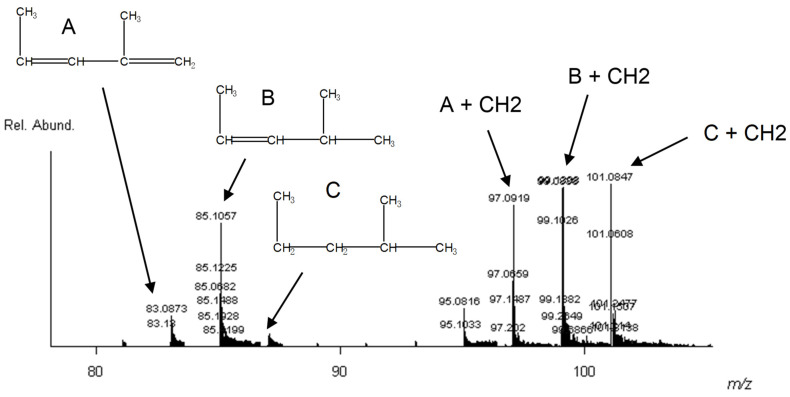
Close up of polypropylene standard mass spectrum (80–110 mass range). All peaks are protonated molecules of the indicated structures.

**Figure 9 polymers-13-02687-f009:**
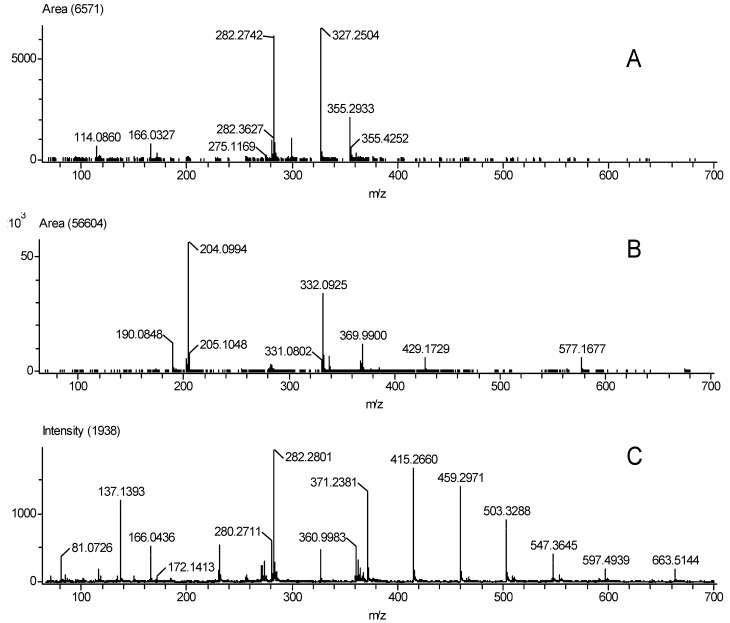
Mass spectra for carpet samples (**A**) 19 (nylon), (**B**) 20 (polyester) and (**C**) 21 (olefin) at 30 V.

**Table 1 polymers-13-02687-t001:** Manufacturer provided melting points of the various polymer standards used in this study (Scientific Polymer Products, Inc., Ontario, NY, USA).

Polymer	Melting Point °C
Nylon 6	220
Nylon 6/6	254
Nylon 6/9	210
Nylon 6/10	217
Nylon 6/12	250–260
Nylon 11	185
Nylon 12	178
PET	252
PBT	225
PCDT	218
Polypropylene	160
Polyethylene	121

PET: Polyethylene terephthalate, PBT: Polybutylene terephthalate, PCDT: Poly (1, 4-Cyclohexylene-Dimethylene terephthalate).

**Table 2 polymers-13-02687-t002:** FTIR spectra and assignment of nylon-66.

Frequency, cm^−1^	Assignment
3182	N-H stretching
3080	C-H asymmetric stretching
3020	C-H symmetric stretching
2841	CH_2_ symmetric stretching
1745	C=O stretching
1660	Amide I band
1541	Amide II band
959	C-C stretching

**Table 3 polymers-13-02687-t003:** Calculated exact masses of the monomers and dimers of nylons and polyesters, as well as their protonated exact masses.

Polymer	M	M+H	2M	2M+H
Nylon 6	113.0839	114.0919	226.1679	227.1759
Nylon 6/6	226.1679	227.1759	452.3361	453.3441
Nylon 6/9	268.2149	269.2229	536.43	537.4380
Nylon 6/10	282.2306	283.2386	564.4613	565.4693
Nylon 6/12	310.2618	311.2698	620.5239	621.5319
Nylon 11	183.1621	184.1701	366.3244	367.3324
Nylon 12	197.1778	198.1858	394.3558	395.3638
PET	192.0421	193.0501	384.0844	385.0924
PCDT	220.0734	221.0814	440.147	441.1550
PBT	274.1205	275.1285	548.2409	549.2489

## Data Availability

Data is contained within the article.

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
