# Peer review of "Forensic Analysis of Polymeric Carpet Fibers Using Direct Analysis in Real Time Coupled to an AccuTOF™ Mass Spectrometer"

_polymers, 2021, doi:10.3390/polym13162687_

Round 1

Reviewer 1 Report

In my point of view, this manuscript needs to express more clearly the idea of the objective and novelty of the study, results are not discussed in detailed. Some other comments are given below 1.- Please revise carefully the whole manuscript, I found several typing errors 2.- Some of the references used in the introduction section are too old, you should use a more actualized references specially for concepts., The use of too old references sometimes is valid to recognized the inventions of some important methodologies or techniques, but for explain concepts and give examples of studies are better to use recent publications 3.- For a better visualization, please separate the "Materials and Methods" section in specific subsections for each test, also, the materials section have to be in a separate paragraph subsection 4.- In "materials sections" add molecular weight and brand, for all reagents add the brand. 5.- I did not found FTIR analysis methodology in "Materials and Methods" section 6.- This manuscript needs more detailed discussions and comparison of the results with other studies 7. Eleven references for a experimental paper is too low 8.- This manuscript needs more scientific content, this manuscript looks more as a technical report form 9.- Please revise the "guide for authors" and published articles in this journal to see examples of the references format 10. I can't see the novelty of the study, I found some reviews about DART-AccuTOF tha is used for forensic and analysis of fibers 11. All spectra analysis needs to detailed, I don´t see an analysis of each of the important peaks or signals in the results section

Author Response

Dear Reviewer, 

Please find the response to some of your valid and useful comments.

Thanks!

Reviewer 2 Report

This study focused on polymeric carpet fibers most encountered at a crime scene, which are nylons, polyesters and olefins. Analysis of the multiple polymer types was done using Direct Analysis in Real Time (DART™) coupled to an Accurate time-of-flight (AccuTOF™) mass spectrometer (MS). The work described in the manuscript of quality warrant publication. So the publication is recommended after revision:
Comments

  • Would you explicitly specify the novelty of your work? What progress against the most recent state-of-the-art similar studies was made? what is the gap to cover.
  • The introduction should be clarified in term of uniqueness and advantage what is the novelty of this work over the previous related work. There are many long sentences should be refined.
  • More profound discussions and comparison with other published works are welcomed.
  • The Manuscript needs thorough revision to improve the text quality and readability of work.
  • Please revise the conclusion. It is too long!

Author Response

Dear reviewer, 

Please find the response to your useful feedback of the manuscript.

Thanks!

Regards

Reviewer 3 Report

The MS is just a short communication, not a paper. The authors did not analysis the FTIR results in detail. 

Author Response

Dear Reviewer,

Please find the response to your useful comments.

Thanks!

Regards

Round 2

Reviewer 1 Report

The authors had attended my recommendations successfully

Reviewer 3 Report

OK